# Contrast-Enhanced Ultrasonography (CEUS) in Imaging of the Reproductive System in Dogs: A Literature Review

**DOI:** 10.3390/ani13101615

**Published:** 2023-05-11

**Authors:** Letizia Sinagra, Riccardo Orlandi, Tiziana Caspanello, Alessandro Troisi, Nicola Maria Iannelli, Emanuela Vallesi, Giorgia Pettina, Paolo Bargellini, Massimo De Majo, Cristiano Boiti, Santo Cristarella, Marco Quartuccio, Angela Polisca

**Affiliations:** 1Department of Veterinary Sciences, University of Messina, Viale Palatucci, 13, 98168 Messina, Italy; lesinagra@unime.it (L.S.); nicola_iannelli@libero.it (N.M.I.); giorgia.pettina@studenti.unime.it (G.P.); mdemajo@unime.it (M.D.M.); scristarella@unime.it (S.C.); marco.quartuccio@unime.it (M.Q.); 2Anicura Tyrus Clinica Veterinaria, Via Bartocci 1G, 05100 Terni, Italy; riccardo.orlandi@anicura.it (R.O.); emanuela.vallesi@anicura.it (E.V.); paolo.bargellini@anicura.it (P.B.); 3School of Biosciences and Veterinary Medicine, University of Camerino, Via Circonvallazione 93/95, 62024 Macerata, Italy; 4Clinica Veterinaria Camagna—VetPartners, Via Fortunato Licandro 13, 89124 Reggio di Calabria, Italy; 5Anicura CMV Clinica Veterinaria, Via G.B. Aguggiari 162, 21100 Varese, Italy; 6Tyrus Science Foundation, Via Bartocci 1G, 05100 Terni, Italy; boiti.cristiano@gmail.com; 7Department of Veterinary Medicine, University of Perugia, Via San Costanzo 4, 06126 Perugia, Italy

**Keywords:** CEUS, contrast-enhanced ultrasound, diagnostic imaging, reproduction, dog, testis, prostate, ovary, uterus, mammary gland

## Abstract

**Simple Summary:**

Contrast-enhanced ultrasound (CEUS) has been widely applied for reproductive imaging in humans and animals. This structured literature review aims to analyze the usefulness of CEUS in canine reproduction. Articles from 1990 to 2022 about CEUS in canine testicles, prostate, uterus, placenta, and mammary glands were searched on PubMed and Scopus. Thirty-six total results were found. The analysis of these works enlightened the usefulness of CEUS in testicular abnormalities and neoplastic lesions, except for characterizing tumors. CEUS in dogs was studied in animal models for human prostatic cancer treatment, while in veterinary medicine it was used to study prostatic vascularization and to assess prostatic diseases, showing good specificity for adenocarcinomas. CEUS differentiated the follicular phases in ovaries. In CEH-pyometra syndrome, it differentiated endometrium and cysts, and highlighted angiogenesis. CEUS was shown to be safe in pregnant dogs and was able to assess normal and abnormal fetal–maternal blood flow and placental dysfunction. In normal mammary glands, CEUS showed vascularization only in diestrus, with differences between mammary glands. CEUS was not specific for neoplastic versus non-neoplastic masses and for benign tumors, except for complex carcinomas and neoplastic vascularization. Works on CEUS showed its usefulness in several pathologies as a non-invasive, reliable diagnostic tool.

**Abstract:**

The use of contrast-enhanced ultrasound (CEUS) has been widely reported for reproductive imaging in humans and animals. This review aims to analyze the utility of CEUS in characterizing canine reproductive physiology and pathologies. In September 2022, a search for articles about CEUS in canine testicles, prostate, uterus, placenta, and mammary glands was conducted on PubMed and Scopus from 1990 to 2022, showing 36 total results. CEUS differentiated testicular abnormalities and neoplastic lesions, but it could not characterize tumors. In prostatic diseases, CEUS in dogs was widely studied in animal models for prostatic cancer treatment. In veterinary medicine, this diagnostic tool could distinguish prostatic adenocarcinomas. In ovaries, CEUS differentiated the follicular phases. In CEH-pyometra syndrome, it showed a different enhancement between endometrium and cysts, and highlighted angiogenesis. CEUS was shown to be safe in pregnant dogs and was able to assess normal and abnormal fetal–maternal blood flow and placental dysfunction. In normal mammary glands, CEUS showed vascularization only in diestrus, with differences between mammary glands. CEUS was not specific for neoplastic versus non-neoplastic masses and for benign tumors, except for complex carcinomas and neoplastic vascularization. Works on CEUS showed its usefulness in a wide spectrum of pathologies of this non-invasive, reliable diagnostic procedure.

## 1. Introduction

Ultrasonography is the best imaging technique for real-time assessment of the reproductive organs and pregnancy in both human and veterinary medicine. Among the most advanced ultrasound techniques developed in recent years, Doppler ultrasonography, elastography, three-dimensional (3D) and four-dimensional (4D) ultrasound, and CEUS are widely applied in reproductive imaging of humans and small animals [1,2,3,4,5,6,7,8,9,10,11,12,13,14,15,16,17,18,19,20,21,22,23,24,25,26,27]. Clinical data confirm that using these new techniques provides additional information concerning the reproductive problems of the evaluated organs and increases the diagnostic potential of conventional ultrasound [12].

CEUS is based on intravenous injection of gas-filled microbubbles that allow real-time ultrasound tracking of the perfusion of tissues by following contrast circulation into the vascular bed. This technique has been widely used for many years in a wide range of clinical settings in thousands of both human and animal patients and, to date, with limited side effects [28,29,30,31,32,33,34,35]. Commonly used ultrasound contrast agents stay in the vessels and never cross the endothelium, neither in mares nor in bitches [36,37]; their bubbles are similar in size to erythrocytes [38] and have not shown any mutagenic or teratogenic effects in vivo [39]. However, studies regarding the use of this functional imaging methodology in human and veterinary obstetrics and gynecology are relatively few compared to the wide range of clinical applications. CEUS has been applied in different reproductive organs such as the prostate, testicles, mammary glands, the uterus, and placenta to better characterize reproductive diseases that might be related to infertility [40,41,42,43,44]. Recently, uterine perfusion in mares during normal pregnancy has been evaluated with CEUS, which did not lead to any evident side effects [36].

The main goal of this review is to systematically analyze the utility of CEUS in several aspects of reproductive imaging, by highlighting the contribution that this technique can add to standard ultrasonographic methods in characterizing reproductive physiology as well as in increasing the diagnostic accuracy of pathologies that may involve testicles, the prostate, the uterus, placenta, and mammary glands. This review will focus on male and female dogs, comparing the use of CEUS in human gynecology and andrology.

## 2. Materials and Methods

We delimited our area of interest in the application of CEUS to the reproductive system of male and female dogs. A search was conducted on the PubMed and Scopus websites in September 2022. The research was carried out by two authors independently, according to a pre-established criterion of combinations of keywords. The combinations of keywords used for the PubMed search are shown in Table 1, while the Scopus advanced search tool was used by inserting the combinations of keywords shown in Table 2, each separated from the other by the operator “AND” and limiting the search area to the veterinary subject {e.g., TITLE-ABS-KEY (“Contrast-enhanced ultrasound” AND pregnancy) AND [LIMIT-TO (SUBJAREA, “VETE”)]}.

Eligibility criteria were works published in indexed journals between 1990 and 2022 and written in the English language, including studies that dealt with the use of CEUS in at least one of the following areas: testis, prostate, ovary, uterus, breasts. Duplicates of the same article, works that did not concern dogs, and articles that concerned the use of CEUS or applied it in organs or systems different from the reproductive ones were excluded. Reviews were also excluded, since these types of papers do not deal with the direct application of this diagnostic technique and report indirect data from other authors (Table 3).

## 3. Results

### 3.1. Analysis and Selection of Studies

The research on PubMed and Scopus yielded 102 and 27 results, respectively. Thirty-six works were duplicate and eliminated, and the remaining 93 were subjected to screening. Fifty-seven of them did not meet the inclusion criteria and were eliminated, while 36 works met the eligibility criteria and were selected for the study (Figure 1).

### 3.2. CEUS and Testis

In human medicine, CEUS could better differentiate testicular lesions from acute scrotal diseases, such as infarction, trauma, torsion, and varicocele as well as neoplastic lesions (both benign and malignant) from non-neoplastic ones [8,45,46,47,48,49,50]. A systematic review and meta-analysis assessed a high diagnostic accuracy of CEUS in characterizing non-neoplastic vs. neoplastic testicular lesions (96%), distinguishing malignant and benign testicular masses in men (96%), and finding high pooled positive and negative predicting values (0.85 and 0.91, respectively) [51]. Few reports have been published in this field on dogs [41,52,53,54].

The first record of the use of CEUS in canine testicular disease was in 1996, in a study on the detection of testicular ischemia [55]. Six male mongrel dogs were sedated, and 0.01–0.025 mL/kg of MRX-115 contrast agent (ImaRx Pharmaceutical, Tucson, AZ, USA) were administered intravenously in three injections at 10-min intervals. Ischemia was induced by surgically ligating the spermatic artery, and images were acquired before and after this procedure and before and after inoculation of MRX-115. Results showed a fast enhancement (30–45 s) that persisted for several minutes, highlighting the ability of CEUS to detect testicular ischemia with good accuracy and a high level of confidence [55].

Also in 1996, Pugh et al. [56] evaluated the feasibility of CEUS using EchoGen (Sonus Pharmaceuticals, Bothell, WA, USA) in several organs, among which the testes, of three healthy beagle dogs. A bolus of 0.01–0.65 mL/kg contrast medium was injected with a protocol of four injections at each session. The testes were much less enhanced than the kidneys or aorta, probably because of the proportionally minor volume of blood flow. CEUS allowed direct evaluation of tissue vascularization, indicating several possible clinical applications and its usefulness in differentiating vascular conditions based upon characteristics of flow, and showing otherwise unreachable vessels [56].

In 1997, six mixed-breed dogs were anesthetized and examined in an investigation into the CEUS assessment of gonadal torsion in an animal model [57]. The contrast agent used was FS069 (Molecular Biosystems, San Diego, CA, USA), first with a bolus of 1.5 mL (approximately 0.075 mL/kg) contrast medium, followed by a saline solution flush to clear the line. Then, once the contrast agent had been washed out from the testes, a 3.0 mL bolus was administered, for a total of 16–42 contrast medium injections in each animal in 5–7 h. The testicles were surgically exposed via a scrotal incision, and the probe was positioned directly on them. Testicular flow was assessed, then the right testis was rotated manually 180 and 360 degrees, while the left one remained in its position as a term of comparison. Torsion was maintained for 4–6 h, and testes were evaluated hourly by gray scale, color flow, duplex Doppler, power Doppler, and CEUS. Before rotation, CEUS showed an enhancement after 20 s from the bolus injections, with an increase for 5–10 s, followed by a gradual decrease; the enhancement duration was about 30 s. After 180-degree torsion, a residual perfusion was visible and persisted until detorsion. After 360-degree rotation, perfusion was fully interrupted, and when detorsion was applied, a marked hyperemia was observed. In any case, the authors did not report how long it took for decreasing or interruption of vascularization [57]. CEUS showed great sensitivity in defining flow, and the authors suggested its use for the diagnosis of testicular ischemia [57].

The assessment of vascularization in testicles with chronic lesions in dogs using SonoVue^®^ (Bracco, Milan, Italy), a contrast agent made of sulfur hexafluoride microbubbles, was first reported by Volta et al. (2014), who injected it in an intravenous bolus of 0.03 mL/kg, followed by a flush of 5 mL of saline solution [52]. The authors considered non-neoplastic and neoplastic lesions, comparing the sub-capsular and intra-parenchymal artery pattern of pathological testes with the physiological testes of healthy dogs. The results showed that soon after contrast agent injection, the normal testis blood flow showed a homogeneous, moderate wash-in enhancement followed by a rapid and homogeneous wash-out. Non-neoplastic lesions (testicular degeneration, atrophy, chronic necrotizing orchitis, and interstitial cell hyperplasia) showed a homogeneous pattern characterized by a lower enhancement than that on normal testes. In contrast, neoplastic testes showed an inhomogeneous pattern with a hyper-enhancement compared to the surrounding normal tissue. However, these authors found no difference between the different types of testicular tumors examined [52]. A specific study on testicular interstitial cell tumors (ICT) was performed by Quartuccio et al. (2018) in non-sedated dogs [54]. The authors described the images obtained with CEUS (0.03–0.04 mL/kg of sulfur hexafluoride microbubbles immediately followed by a 5 mL saline flush) compared to B-mode and color Doppler ultrasound. ICT was characterized by inhomogeneous and higher enhancement compared to the surrounding tissue, an enhanced rim, and prominent inner vessels in line with previous results described by Volta et al. (2014) [51]. However, since there were higher perfusion values than the ones in Volta et al. [51], the authors suggested a potential role of sedation as a possible variability factor in the CEUS studies [54].

Testicular CEUS evaluations after a calcium chloride sterilant injection were reported in 2019 [58]. CEUS was performed using SonoVue^®^ (Bracco Imaging, Italy) before (T_0_) and 5 months after (T_5_) intratesticular and or epididymal injection of CaCl_2_, and testicular morphological changes were studied. Scrotal CEUS at T_5_ showed reduced perfusion in the areas injected with CaCl_2_ compared with the same testicle imaged at T_0_, independently of the approach (intra-testicular vs. intra-epidydimal injection) used; the presence of anechoic zones in testicular and epididymal parenchyma represented tissue damage caused by CaCl_2_. This substance exerted a necrotizing effect on the germinal epithelium, confirmed by the CEUS visualization of a reduced blood perfusion [58].

Orlandi et al. (2022) performed qualitative and quantitative evaluations of contrast enhancement patterns in 27 dogs presenting 45 testicular masses diagnosed by histopathology following orchiectomy [41]. CEUS was performed by vein injection of a bolus of 0.03 mL/kg of SonoVue^®^ (Bracco Imaging, Milan, Italy), followed by a 5 mL saline flush. Interestingly, a high proportion of Leydig tumors showed perilesional vessels compared to other testicular tumors that had an intralesional vascularization pattern. Although the authors did not mention specific cut-off values, they found that in leydigomas and seminomas, peak intensity and area under the curve were higher, and time to peak was lower compared to surrounding normal parenchyma. In any case, these quantitative parameters did not vary among the different tumor types. Nevertheless, CEUS identified all tumors, including a sertolioma in a retained testicle that escaped color Doppler and B-Flow imaging [41]. Thus, the authors concluded that while CEUS was highly effective in detecting testicular tumors, it added no contribution to their characterization [41].

Recently, CEUS has been used to characterize the pattern of canine testicular blood flow after chemical sterilization with CaCl_2_ injection in alcohol, showing an alteration of the vascularization compared with normal testes [53]. This sterilization treatment induced both faster wash-in and wash-out phases than in control testes, and presented a mostly anechoic pattern, with a few hyperechoic vascular focal spots [53].

### 3.3. CEUS and Prostates

#### 3.3.1. CEUS in Canine Prostates as an Animal Model

In human medicine, several papers and reviews have been published on the clinical application of CEUS in the discovery, localization, and assessment of treatment for prostate cancer [3,59,60,61,62,63,64,65,66,67].

Forsberg et al. [68] evaluated transrectal CEUS in a canine prostate cancer model, by implanting canine-transmissible venereal sarcoma (CTVS) cells into the prostate of 24 dogs and then evaluating them by ultrasound at different times (15, 18, 21, 24, and 28 days after tumoral cell inoculation). The protocol for CEUS required IV administration at 1 mL/s of 2–5 injections of a 1% solution of Sonazoid, in dosages of 0.00625 to 0.20 μL of microbubble/kg, at intervals of 10–15 min between injections. Characterization of the prostatic lesions (number, size, location) was determined by transrectal ultrasonography (TRUS) and CEUS. Finally, the dogs were euthanized, and gross histologic examination of the prostate and local lymph nodes was performed. CEUS enabled the individuation of tumors inside the parenchyma and outside the capsule of the prostate, in the rectal wall and close lymph nodes; also, Sonazoid allowed a better visualization of the vascularity of prostatic tumor and delineation of the size and shape of the mass. Indeed, tumors were characterized by tortuous intra-tumoral vessels, with avascular central regions associated with necrosis. Histopathologic examination confirmed the ultrasonographic findings. CEUS was more accurate than conventional TRUS in diagnosing prostate tumors in dogs and allowed the reliable identification of 5-mm tumors. Contrast-enhanced power Doppler TRUS could be useful in showing focal prostatic abnormalities, in providing additional biopsy guidance, and in allowing higher cancer detection rates [68]. In any case, the authors did not report any quantitative analysis but only the results of the qualitative evaluation of images through a comparison of CEUS and B-mode images.

In many studies, CEUS for canine prostates was used for monitoring prostatic thermal therapy of prostatic carcinoma in a canine animal model [22,69,70,71,72,73]. It consists in applying high temperatures to neoplastic tissues, which denature and coagulate structural proteins and blood supply, hence determining the death of cells. Tissues can be heated by local application of microwave antennas, radiofrequency ablation (RFA) electrodes, laser fiber optic probes, or high-intensity focused ultrasound transducers [70].

Among these techniques, prostate RFA is considered a feasible treatment for prostate cancer [69,72,73]. CEUS has been used as guidance to monitor and control the procedure [22], to analyze feasibility and possible ways to improve the technique [69,71], and to evaluate the outcome of RFA of canine prostate lesions [73]. In all these cases, CEUS confirmed its efficacy. Even a comparation between B-mode US, CEUS, and MRI for the study of RFA lesions showed similar results between CEUS and MRI; however, the former has the advantage of being more economical and using more convenient equipment and faster scanning, thus representing the best choice [72].

In 2006, CEUS was used as guidance to monitor and control radio frequency (RF) ablation of a canine prostate in an animal model [22], with a protocol of a preoperative bolus injection of Sonazoid^®^ (0.04 mL/kg) followed by a 5 mL saline flush, a continuous infusion (0.015 μL/kg at 11 mL/min) during the RF ablation procedure, and then other bolus injections after completing each RF ablation and at the end of the entire ablation procedure. Pulse inversed harmonic (PIHI) CEUS and Doppler CEUS were performed with a transrectal approach. CEUS enabled the visualization of both the normal prostate and the thermal lesions created. PIHI CEUS better differentiated thermal lesions, while power Doppler CEUS showed blooming artifacts that covered thermal lesion areas, hindering their vision. Measurements obtained from contrast-enhanced PIHI had good agreement and good linear correlation with pathological findings. CEUS also allowed the direct visualization of the urethra and neurovascular bundles, minimizing the possibility of damaging these areas [22].

In a similar study, Liu et al. [69] performed CEUS-guided RFA of a canine prostate with application of urethral and neurovascular bundle (NVB) cooling. Study groups received trans-urethral infusion of cold saline solution, control groups did not. During ablation, transrectal PIHI CEUS was performed, after an intravenous bolus injection (0.04 mL/kg) and infusion (0.015 L/kg/min) of Sonazoid^®^. PIHI CEUS allowed visualization and monitoring of urethral and NVB blood flow during the ablation. Contrast-enhanced US could successfully guide RF ablation of the entire prostate. In control groups, US and pathology showed damage to the urethra and the NVB, confirming the efficacy of cold in protecting these tissues during this procedure [69].

In another animal model, CEUS was used in association with MRI for monitoring prostate microwave focal thermal therapy [70]. Seven beagle dogs underwent CEUS and MRI scans before, during, and after microwave heating of the prostate. CEUS was performed by an injection in bolus of Definity^®^ (0.3 mL, then 10 mL saline flush). These exams aimed to accurately evaluate the thermal lesion and to individuate areas of vascular compromission inside the lesion, by comparison with the surrounding normal tissue. The authors found that these techniques may possibly provide indication of damage during and right after the procedure, within 2 h. However, the injury was evident over time after therapy, starting with a peripheric hyperemic rim, which can become necrotic and is therefore a better predictor of damage [70].

Hu et al. [71] used RFA on healthy canine prostates to create one lesion in each lobe. An amount equal to 2.4 mL of SonoVue boluses were infused, followed by 5 mL of physiologic saline, thus imaging was suddenly performed; immediately after the procedure, all dogs were euthanized to remove the prostate and compare the gross pathology and the histopathology. CEUS was able to detect thermal lesions as areas without vascularization, hypoechoic to surrounding normal parenchyma, with a more precise demarcation than B mode US [71].

Feng et al. [72] performed CEUS using a bolus of 2.4 mL of the same medium contrast, followed by a 5 mL saline flush to evaluate RFA lesions at different periods of time (7, 30, 90, and 180 days) after the RFA procedure and comparing pathologic results after the dogs’ euthanasia [72] Finally, Jia et al. [73] used the same technique to create thermal lesion in healthy canine prostates, then they performed B-mode US and CEUS immediately after, 1 week after, and 1 month after [73]. The medium contrast (SonoVue) was injected using a 2.4 mL bolus in the forelimb vein, followed by a 5 mL flush of saline solution. After each evaluation, the dogs were euthanized to compare pathologic results. In the three cases, the authors showed a correspondence between the study of lesions by CEUS and the pathologic results, due to the possibility to follow the dynamic process of the range of lesions and the vascularization changes induced by RFA [71,72,73].

#### 3.3.2. CEUS in Canine Prostatic Evaluation in Veterinary Medicine

A study of prostatic vascularization through the CEUS technique in veterinary medicine was first published in 2001 by Krüger Hagen et al. [74] using perfluorobutane microbubbles of Sonazoid^®^ (GE HealthCare), a first-generation contrast agent. The contrast medium was infused through an 18-G angiocatheter placed in a forelimb vein [74]. The first dog received doses ranging from 0.0125 to 0.0375 mL microbubbles/kg, and the other dogs received mainly doses of 0.00625 mL micro-bubbles/kg. All dogs received, on average, six injections of Sonazoid. Every injection of Sonazoid was followed by a 10 mL flush of saline solution [74]. The authors evaluated prostatic blood flow in five mongrels under general anesthesia. This study compared 3D and 4D ultrasound before and after contrast agent injection, obtaining a better visualization of prostatic blood flow [74].

In 2009, Russo et al. [75] used a second-generation contrast agent (SonoVue, Bracco Imaging, Milan, Italy) to study the blood flow of normal canine prostates in five anesthetized dogs. After injection (a bolus of 0.03 mL/kg of SonoVue^®^, followed by a bolus of 5 mL of saline solution), prostatic artery branches were visible in 10–15 s, then the contrast agent spread into the prostatic parenchyma from the dorsolateral surface with a homogeneous enhancement and wash-out phase. Moreover, after SonoVue injection, a stronger Doppler signal compared to the pre-contrast evaluation was evidenced [75]. In 2011, Vignoli et al. [32] compared perfusion peak intensity (PPI) and time to peak (TTP) values between normal and pathological prostates in dogs. SonoVue^®^ was infused into the cephalic vein at a dose of 0.03 mL/kg, followed by a rapid bolus of 5 mL of saline solution. Through the study of different pathologies (successively confirmed by histological examination), the authors found three different range of values: normal prostate for the dogs with benign prostatic hyperplasia and mixed benign lesions; lower values in dogs with prostatitis and leiomyosarcoma; higher values in dogs with adenocarcinoma. Therefore, the authors concluded that it was not possible to distinguish the different types of prostatic diseases using only the CEUS technique, except for adenocarcinoma [32]. Similar results were obtained by Russo et al. (2012) [40]. Both studies were conducted in dogs under total anesthesia, differently from the study of Troisi et al. (2015) [30], who used contrast agent (a bolus of 0.03 mL/kg of SonoVue^®^ followed by a rapid bolus of 5 mL of saline solution) in non-anesthetized animals [30]. Based on their results, benign prostatic hyperplasia can be clinically recognized by a chaotic vascular pattern during the wash-in phase, probably due to simultaneous blood vessel enhancement and the presence of circular avascular areas characterizing prostatic cavitation [30]. Similar findings were reported in dogs with prostatitis: in addition to the chaotic vascular pattern during the wash-in phase, there was a significative enhancement during the wash-out phase of the vessels encircling the urethra compared to the surrounding parenchyma [30]. Concerning specific vascular patterns of prostatic tumors, adenocarcinoma showed large arteries with a chaotic appearance instead of the normal subcapsular pattern and a general hypoechoic, nodular vascular pattern of the tumors compared to the parenchyma. Lymphoma showed an increased enhancement during the early wash-in phases and a heterogeneous hypoechoic appearance with fine echo pollution of the microcirculation during the wash-out phase [30].

A similar study was conducted by Bigliardi and Ferrari, in 2010 [76], using a different contrast medium (Levovist, injected by a bolus in the cephalic vein through an 18G catheter at a dose 300 mg/mL, followed by a 15 mL saline flush) [76]. All dogs were premedicated with atropine sulfate and acepromazine, then anesthetized by isoflurane administered through a face mask. The authors classified prostatic vascularization as poor, moderate, and good depending on visualized vessels: only prostatic arteries on the dorsolateral surface, capsular vessels under prostatic capsule in close proximity to it, intraprostatic arteries, the small vessels of the parenchyma, and periurethral vessels, respectively [76]. In this study, CEUS allowed for the significant improvement of the visualization of prostatic blood flow. Moreover, in prostatic arteries, mean systolic and peak velocity were 38.5 m/s and 15.7 m/s, respectively; mean end-diastolic velocity was 5.9 m/s; the resistive index and pulsatility index values were 0.83 and 2.3, respectively. In capsular arteries, mean systolic and peak velocity were 12.05 m/s and 9.2 m/s, respectively; mean end-diastolic velocity was 4.2 m/s; the resistive index and pulsatility index values were 0.7 and 1.31, respectively [76].

Other studies focused the use of CEUS on the prostate of castrated dogs. In 2020, Yoon et al. [77] compared the use of ultrasound and CT contrast agent (dose of 0.125 mL/kg via the cephalic vein using a 3-way stopcock and 20-gauge catheter, followed by a bolus of 5 mL of saline) in dogs with normal prostates against those with benign hypertrophic prostates and castrated dogs. The results of normal and hypertrophic prostates were in line with previously described works [30,32,58,76]. In castrated dogs, the authors found similar enhancement patterns 15 and 30 days after castration and slower wash-in and wash-out phases associated with a marked peak intensity, leading to a significantly reduced peak intensity 60 days after surgery [77]. Another study focused on the follow-up of prostatic blood flow in castrated dogs for at least 6 months, using a bolus of 0.03 mL/kg of SonoVue^®^ followed by a rapid bolus of 5 mL of saline solution [78]. In contrast to the previous work [77], the authors found a significantly higher PPI in castrated dogs than in intact dogs and a similar TTP [78].

### 3.4. CEUS and Ovaries

Until now, only a few research studies have investigated the efficacy of CEUS in studying ovarian vascularization: the behavior of ovarian lesions (tumors or tumor-like) in women [79], functional ovarian vascular changes induced by GnRH-analogue administration in a murine model [80], and in other experimental conditions in rhesus monkeys [81].

In their animal model of 1997, Brown et al. [57] also applied CEUS for the evaluation of gonadal torsion in three female dogs by exposing the ovaries and applying the same protocol as in male dogs. Prior to manipulation, CEUS showed significant enhancement of ovarian perfusion, which led to an increase in strength and amplitude of the signal. In 180- and 360-degree rotated ovaries, CEUS did not show any visible enhancement, reflecting profound ischemia. After release, perfusion returned to normal. As in male dogs, CEUS proved its usefulness in evaluating ovarian perfusion and assessing ischemia [57].

On the other hand, there is only one recent paper about CEUS and ovaries in veterinary medicine, by Nogueira Aires et al. in 2022 [82]. SonoVue^®^ was injected via IV (0.01 mL/kg), followed by 5 mL of saline solution (NaCl 0.9%). The authors studied the ovarian vascularization pattern during the follicular phase and the early luteal phase, comparing the results to the increasing progesterone levels [82]. The authors verified peak contrast intensity (PPI peak in % of pixels), time to peak (TTP), mean transit time (MTT), area under the curve (AUC), and the average number of pixels. Results showed that the enhancement, at first minimal during early proestrus, became more evident during the progressing of estrus until the post-ovulatory period. PPI, AUC, and MTT were significantly higher in the last evaluation, while TTP and pixels remained unchanged during all evaluations [82].

In conclusion, CEUS can help monitor the increase in vascularization during the follicular and post-ovulatory phases of bitches, even if these values are not closely related to increasing progesterone levels [82].

### 3.5. CEUS and the Uterus

The evaluation of endometrial and subendometrial vascularization by CEUS in women with infertility or different uterine pathologies such as endometrial carcinoma, endometrial hyperplasia, endometrial cysts, and uterine fibroids are quite scarce [14,17,83,84,85,86,87,88].

Indeed, the same paucity applies to the clinical applications of this technique in the gynecology of small animals. Indeed, the usefulness of CEUS in uterine examination has only been evaluated by Quartuccio et al. (2020) [42] in a preliminary study. The authors used CEUS, performed using a bolus of 0.03–0.04 mL/kg of SonoVue^®^, instantly followed by a 5 mL flush of saline solution, in dogs with cystic endometrial hyperplasia–pyometra complex (CEH) to assess the vascularization in endometrial micro-vessels and to evaluate the feasibility of this technique for the description of the typical lesions of this syndrome [42].

The authors performed qualitative and quantitative analyses of the contrast enhancement pattern and perfusion. From 6–8 s after contrast agent injection, the hyperplastic endometrium showed a homogeneous enhancement (wash-in phase), then the decrease in enhancement was slow during the wash-out phase, with evidence of contrast up to 2 min. The examination evidenced the opposition between the intense enhancement in the hyperplastic endometrium and the lack of signal inside the cysts (Figure 1). With regard to the vascular pattern, peripheral vessels were well represented, unlike vessels in the thin muscle layer, where only the small perpendicular ones were evident [42]. CEUS showed a rapid enhancement of endometrial proliferation in the uterus of all bitches correlated with an inflammatory condition, confirmed by a positive CD34 immunostaining in post-hysterectomy histological examination [42]. The CEUS investigation accurately highlighted the high number of vessels generated by the local angiogenesis, as demonstrated by immunohistochemical examination. Limitations of this work are the number of subjects and the inclusion criteria (high grade of CEH-pyometra) without a comparison with other grades of the pathology and other females evaluated during a non-pathologic diestrus phase [42].

### 3.6. CEUS in Pregnant Females

In human medicine, contrast-enhanced ultrasound is not currently used for clinical practice, but only for research reasons, as the required contrast agents have not yet been approved for use in pregnant women, even though their safety has already been demonstrated [89,90].

CEUS was used for qualitative and quantitative evaluation of blood flow parameters of fetal–maternal circulation at the 23rd, 30th, and 45th day of pregnancy in nine healthy pregnant dogs by Orlandi et al. (2019) [37], using 0,03 mL/kg of SonoVue^®^ contrast agent, followed by a 5 mL saline flush. Independently of the gestational days examined, the wash-in phase was characterized by a first appearance of the contrast agent in the uterine artery, followed by fast distribution in the vessels of the uterine wall [37]. Then, the contrast agent distributed radially from the peripheric parietal vessels to the placental vessels. No contrast agent was evidenced in embryos or fetuses. At day 45 of gestation, a 20% smaller placental area was detected on CEUS images compared with B-mode images, probably due to the presence of marginal hematomas that were not enhanced [37].

Quantitative analysis was performed on time-intensity curves of contrast-enhancement obtained by ROIs positioned within proximal and distal placenta and in the uterine artery [37]. Independently of gestational days, the parameters evaluated did not vary between the ROIs, but all parameters were lower in the uterine artery. The area under the curve (AUC) values did not change in the placental ROIs, but in the uterine artery they were significantly lower at day 30 than at day 23 [37]. The authors interpreted this result as a possible consequence of trophoblast invasion and changes in the conformation of arteries and capillary course. Since the CEUS time-intensity curve parameters did not differ between the proximal and distal placenta, the authors suggested that the evaluation of distal placental enhancement could be left out in further studies [37]. This work demonstrated that CEUS can be safely used in pregnant dogs. No adverse effects related to the presence of the contrast agent in the fetal–maternal circulation were noted, as indirectly proved by the absence of any visible enhancement in the embryos or fetuses and by the fact that all the bitches evaluated gave birth to viable pups without abnormalities. Thus, CEUS can accurately assess fetal–maternal blood flow at least during the first two-thirds of gestation in dogs; this imaging technique should also be useful in diagnosing placental pathological conditions that likely cause abnormal placental blood flow [37]. As frequently found in this type of work, the main limitation of this study was the low number of cases examined. Therefore, further studies on larger numbers of dogs of different breeds and stages of pregnancy are required to obtain reliable standard parameters that can be safely used as a paradigm to distinguish normal and abnormal pregnancies [37].

More recently, Silva et al. (2021) [43] evaluated the same parameters in 30 pregnant brachycephalic bitches, of which 22 had normal pregnancies and 8 had gestational abnormalities (2 hydrocephalus and 6 anasarca). They performed B-mode, Doppler US, and CEUS using 0.01 mL/kg of SonoVue and 5 mL of saline solution injection on days 25, 45, and 58 of pregnancy, respectively, and collected data on the two fetuses closest to the ovaries [43]. The enhancement pattern showed a diffusion of the contrast medium from outer to inner uterine vessels of the placenta (wash-in phase), then in the opposite direction and a homogeneous decreasing (wash-out phase). Interestingly, the contrast enhancements of abnormal placentas were heterogeneous and less marked, showing an alteration of the vascularization and consequently placental dysfunction. Notably, B-mode and Doppler parameters did not detect significant differences between normal and abnormal placentas, whereas time to peak (TTP), average transmission in time (in sec) (TmT), and AUC evaluated using CEUS were significantly higher in the placentas of bitches with gestational abnormalities [43].

While Doppler fluximetric evaluation of the umbilical cord showed a significant increase in blood flow from day 45 to day 58, placental CEUS quantitative parameters of brachycephalic bitches remained the same throughout pregnancy, suggesting that placental perfusion did not change despite the significant increase in umbilical blood flow [43]. According to Silva et al. (2021), based on CEUS analysis of qualitative and quantitative features, placental perfusion remains constant throughout the whole gestational process in healthy bitches [43]. In bitches with abnormal pregnancies, due to fetuses with anasarca and hydrocephaly, CEUS enables the detection of placental dysfunction characterized by heterogenicity in tissue perfusion, a lower intensity of placental tissue filling, and a delay in perfusion times with a diagnostic accuracy close to 75% [43].

### 3.7. CEUS in Mammary Glands

To the best of our knowledge, there is only one study on the use of CEUS for the analysis of blood flow changes in normal canine mammary glands during the estrous cycle phases [91], but none in those of humans. The authors performed the US imaging on the right mammary chain and the respective superficial inguinal lymph node of six healthy beagles at five time points during proestrus, estrus, early and late diestrus, and anestrus. The contrast medium (SonoVue^®^, Bracco, Milan, Italy) was administered via a 0.04 mL/kg bolus injected into the cephalic vein using a 1 mL syringe, immediately followed by a bolus injection of 2 mL saline bolus. Qualitatively, the mammary glands had a heterogeneous enhancement pattern [91]. The feeding vessels were observed only during the phase of diestrus. Within the same set point of the estrous cycle, no differences in time-intensity quantitative parameters were found between the single mammary glands [91]. The contrast medium (SonoVue^®^, Bracco, Milan, Italy) was administered via a 0.04 mL/kg bolus injected into the cephalic vein using a 1 mL syringe, followed by a 2 mL saline bolus.

However, Vanderperren et al. (2018) observed some differences within mammary glands of each bitch in the different estrous cycle stages [91]. The study of the cranial abdominal mammary gland showed an increase in the AUC from estrus to late diestrus and a decrease from late diestrus to anestrus, with a maximum blood volume in this mammary gland during the period of late diestrus [91]. A similar pattern was also observed in the other mammary glands but was considered not significant [91]. Interestingly, the study of the inguinal gland showed a TTP that was longer during anestrus compared to estrus, suggesting slower blood flow during anestrus [91]. The authors suggested that this result might be related to the lack of hormones such as estrogen and progesterone [91].

### 3.8. CEUS in Mammary Gland Tumors

There are hundreds of studies, including papers, reviews, and meta-analyses dealing with the application of CEUS in the diagnosis of breast tumors in women, which is one of the most diffuse oncological pathologies worldwide [92,93,94,95,96,97,98,99,100].

There are only a few papers regarding the application of CEUS related to canine mammary tumors. Irimie et al. (2016) described the use of CEUS in a case report of a mammary tumor in a 5-year-old silky terrier [101], following the administration of 1 mL/10 kg of SonoVue contrast. The lesion was successively identified by immunohistochemistry as a simple cystic papillary carcinoma with a malignancy grade 2 [101]. Changes in intensity and quantitative parameters (baseline intensity (BI), arrival time (TI), peak intensity (PI), rise time (RT), wash-in, and wash-out) were analyzed as time-intensity curves on three regions of interest (ROIs). CEUS showed a layout of intratumoral microvessels into neoplastic parenchyma. Qualitatively, the enhancement was centripetal, with a heterogeneous distribution of the contrast agent. There were no perfusion defects, and margins were well defined. At peak intensity, the tumor was isoenhanced compared to the normal breast parenchyma [101]. According to a study by Du et al. (2008) [102], who divided the intensity curves into four types in accordance with the slopes of wash-in and wash-out phases, the authors defined the time-intensity curve they obtained as a “type 2” curve, characterized by a “slow in” and “fast out” trend. In other words, the pattern described by this curve consisted in a rather slow enhancement of the tumor lesion in the wash-in phase, followed by a fast wash-out phase, probably due to the scarce presence of vessels. The presence of microvessels was analyzed through a comparison to a study by Gal et al. [103]. It showed a significant correlation between tumor malignancy grade and microvessel density, highlighting that the wash-in and peak intensity were higher in malignant tumors [103]. However, in this report, tumor histopathology showed a moderate malignancy grade, so the authors stated that CEUS could not clearly predict malignancy in their specific case [101].

In 2017, Feliciano et al. [44] compared the effectiveness of B-mode, Doppler, CEUS, and acoustic radiation force impulse elastography in predicting the malignancy of mammary tumors in 153 bitches in which at least one mass was detected. In total, 300 masses were evaluated with different imaging techniques and then submitted to histopathological classification. CEUS analysis was performed with a protocol involving an intravenous 0.1 mL bolus of contrast agent (SonoVue^®^, Bracco, Milan, Italy) followed by a 5 mL saline flush [44]. Images were quantitatively analyzed to obtain perfusion time through the subsequent parameters: wash-in time (WI), time to enhancement peak (TP), and wash-out time (WO) [44]. Qualitatively, the presence or absence of contrast in the tumor was highlighted, and the enhancement pattern was described by evaluating enhancement localization and enhancement of the surrounding normal mammary tissue, pattern, internal homogeneity and perfusion type. CEUS allowed for the evaluation of macro- and microcirculation of the mammary tumors [44]. However, none of the CEUS parameters considered had a significant correlation to mammary mass malignancy, resulting in high sensitivity (80.2%) but low specificity (16.7%) [44]. Nevertheless, the authors noticed a proportionally higher degree of contrast enhancement in the benign masses, as previously reported by Wan et al. (2012) [104]. Thus, CEUS proved to be ineffective in the differentiation of canine mammary tumors, although it was very useful in identifying neoplastic macro and micro vascularization [44]. Conversely, recent systematic reviews and meta-analysis in women showed that CEUS is a good tool for improving ultrasound in differentiating benign and malignant breast lesions [100,105], as well as predicting pathological response in patients with breast cancer and NAC [95].

In a secondary prospective observational cohort study, Feliciano et al. (2018) applied the same imaging techniques to evaluate their accuracy in predicting histopathological classification of 246 canine mammary carcinomas [106]. Among the several parameters evaluated using B-mode, Doppler, CEUS, and elastography, only the width-to-length ratio measured by B-mode ultrasonography and perfusion times by CEUS were significantly different between carcinoma types and/or grades [106].

Contrast (SonoVue^®^, Bracco, Milan, Italy, administered via an intravenous bolus 0.1 mL, followed by a 5 mL saline flush via a catheter in the cephalic vein) made it possible to obtain a prediction of complex-type carcinoma with 62% sensitivity and 60% specificity. Quantitative parameters reported as cut-offs were wash-in and peak enhancement times lower than 7.5 and 13.5 s, respectively, identifying this type of mass with a moderate accuracy [106]. Contrast wash-in, peak enhancement, and washout times greater than 6.5, 12.5, and 64.5 s, respectively, were indicative of grade II and III carcinomas with 68% sensitivity and 62% specificity; thus, moderate accuracy was also shown in distinguishing grade II and III carcinomas [106].

In another secondary prospective observational cohort study, Gasser et al. (2018) evaluated the applicability of different imaging techniques such as B-mode, Doppler, acoustic radiation force impulse elastography, and CEUS (performed via an intravenous bolus 0.1 mL of SonoVue^®^ followed by a 5 mL saline flush, via a catheter in the cephalic vein) examinations in the differentiation of mammary lesions in bitches (neoplastic and benign non-neoplastic) [107]. The authors analyzed data from a previous study by Feliciano et al. (2017) [44], who enrolled 36 mammary lesions and evaluated them physically and through different ultrasound examinations before mastectomy and histopathological classification. Of the 36 benign mammary lesions evaluated, 25 were classified as neoplastic and 11 as non-neoplastic. Qualitative and quantitative parameters evaluated using various ultrasonographic methods were not statistically significant in distinguishing neoplastic from non-neoplastic benign mammary lesions. The quantitative ultrasonographic parameters were then compared between the different types of breast lesions, and again no significant differences were found for the variables studied with Doppler, CEUS, and elastography [107].

In conclusion, according to Gasser et al. (2018), ultrasonographic evaluation of benign mammary lesions in bitches did not allow for differentiation between neoplastic and non-neoplastic masses and showed limited efficacy for the comparison of benign histopathological types [107].

## 4. Conclusions

Nowadays, CEUS is a useful diagnostic tool, widely used in human medicine and, in recent years, increasingly in veterinary medicine. An important difference can be found between the papers found in a specific bibliographic search for the male and female reproductive system, as can be highlighted comparing the results with the keywords “CEUS and prostate” or “CEUS and testis” and those found using keywords such as “CEUS and ovary, CEUS and uterus or CEUS and pregnancy”, except for “CEUS and mammary”. This marked variability probably reflects the incidence of different diseases of varying severity affecting each organ and the varying reliability of CEUS as a non-invasive diagnostic tool in distinguishing these pathologies in relation to well-established and long-standing diagnostic protocols.

This review has some limitations. Despite the structured design of the methodology of research, a complete systematic analysis and comparison of the results could not be performed because of the relatively low number of works evaluated and their heterogeneity (e.g., study design, protocols adopted, type and dosage of contrast agent used), which prevented us from obtaining pooled data. Thus, we chose to limit our work to a report of the current published literature.

Overall, the large difference between the clinical applications of CEUS in reproduction reported for humans (in the order of hundreds) compared to those reported for dogs (in the order of dozens) probably reflects the differential pressure from a wide spectrum of serious and widespread pathologies for a non-invasive, accessible, and reliable diagnostic procedure.

## Data Availability

The data presented in this study are available on justified request from the corresponding author.

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
