# Peer review of "Contrast-Enhanced Ultrasonography (CEUS) in Imaging of the Reproductive System in Dogs: A Literature Review"

_animals, 2023, doi:10.3390/ani13101615_

Round 1
Reviewer 1 Report
The authors present a review on Contrast-enhanced ultrasonography (CEUS) in imaging of the reproductive system in dogs.
CEUS is an underexplored tool in animals. This paper has potentially great value in the veterinary field. However, before accepting the manuscript some concerns need to be addressed.
Title: What does structured review mean? Why didn't the authors adhere to PRISMA guidelines to make a systematic review? Otherwise, is this a narrative review? Please specify this aspect.
Introduction: I would delete the following sentence since it sounds non-scientific "The reader can 66 find more information on general issues and technical details by consulting some of the 67 several reviews dedicated to CEUS"
Results: Please change the sentence "The research on PubMed and Scopus gave, respectively, 102 and 27 results (129 total 114 results)" For example, you can write "The research on PubMed and Scopus gave 102 and 27 results, respectively.
Paragraph on CEUS and testis cancer you should include level 1 evidence studies, I suggest:
Tufano, A.; Flammia, R.S.; Antonelli, L.; Minelli, R.; Franco, G.; Leonardo, C.; Cantisani, V. The Value of Contrast-Enhanced Ultrasound (CEUS) in Differentiating Testicular Masses: A Systematic Review and Meta-Analysis. Appl. Sci. 2021, 11, 8990. https://doi.org/10.3390/app11198990.
Are there limitations in this review? please specify if yes.
Conclusion: please delete "conclusion and remarks" and just write Conclusion and be more concise.
I Thank the authors for providing this manuscript.
Reviewer 2 Report
The review is well-structured. In my opinion, this review is of interest and may become suitable for publication after some minor modifications/clarifications.
Line 26-29: From “CEUS to adenocarcinomas”. It is not clear what you are trying to state… I suggest rephrasing it.
Line 35: “Cheap diagnostic tool”. I do not agree with this statement. The value of a bottle of contrast medium is 100-150 euros / 100-150 pounds. The contrast procedure is not cheap but quite expensive for the owner. I suggest removing the previous statement.
Line 51-52: “inexpensive” same as mentioned above.
Line 66-68: From “The reader to CEUS” It is not relevant. I suggest removing it.
Line 134: add the reference.
Line 143: add the reference.
Line 154-158: it would be interesting to know if they reported a specific time when the vascularisation started to decrease or was no longer visible in both the 180 and 360°. I suggest adding this if the study mentions it.
Line 161: add the reference.
Line 175: add the reference.
Line 183: add the reference.
Line 200-203: How did they differ these parameters from the normal parenchyma did they mention a cut-off value or if one of those parameters was more sensitive compared to the other to differentiate neoplasia from non-neoplastic parenchyma? If so, I suggest adding these pieces of information.
Line 203: add the reference.
Line 203-205: is this your statement or the above-mentioned author's statement if so, add the reference here.
Line 211: “Comparing” is not the correct verbal form.
Line 214: add the reference. From now on I am not going to write anymore to add the reference at the end of each paragraph/ sentence after mentioning a paper. Check carefully and correct accordingly.
Line 220-357: Justify text.
Line 228-230: Same as the comment on lines 200-203.
Line 281: if reported in the paper I suggest mentioning roughly the amount of time.
Line 492-494: I suggest mentioning the acronym “a”, “t”, and “triangle” in the text and explaining them.
Line 539: “Feeding” change capital F.
Line 663: as mentioned above for the abstract and the summary. I suggest removing the statement “inexpensive”.
